# Golgi Apparatus Regulates Plasma Membrane Composition and Function

**DOI:** 10.3390/cells11030368

**Published:** 2022-01-22

**Authors:** Ilenia Agliarulo, Seetharaman Parashuraman

**Affiliations:** Istituto di Endocrinologia e Oncologia Sperimentale (SS), Consiglio Nazionale delle Ricerche, 80131 Napoli, Italy

**Keywords:** Golgi apparatus, glycosylation, palmitoylation, TGN sorting, membrane contact sites

## Abstract

Golgi apparatus is the central component of the mammalian secretory pathway and it regulates the biosynthesis of the plasma membrane through three distinct but interacting processes: (a) processing of protein and lipid cargoes; (b) creation of a sharp transition in membrane lipid composition by non-vesicular transport of lipids; and (c) vesicular sorting of proteins and lipids at the trans-Golgi network to target them to appropriate compartments. We discuss the molecules involved in these processes and their importance in physiology and development. We also discuss how mutations in these molecules affect plasma membrane composition and signaling leading to genetic diseases and cancer.

## 1. Introduction

The plasma membrane (PM) plays an important role in being the interface of a cell with its environment. Receptors localized to the PM sense changes in the environment and transmit signals to the inside of the cells so as to appropriately modify the state of the cell. The PM interface not only receives signals but also sends them by exposing several important molecules on the cell surface, which advertise the cell identity and state to the extracellular environment. This bidirectional signaling has important roles in both unicellular and multicellular organisms promoting cell division and differentiation, cell migration, cell death, infection, etc. Given the enormous importance of PM signaling, understanding the processes that contribute to achieving and maintaining appropriate PM composition in terms of lipids and proteins is vital. Secretory pathway processes play an important role in this by synthesizing the PM localized molecules and transporting and targeting them to appropriate membrane domains. The endocytic pathway also plays an important role in regulating the levels and localization of the PM receptors. This has been extensively reviewed elsewhere [1,2,3,4]. The review here will focus on the role of the secretory pathway, especially the Golgi apparatus, in achieving the appropriate PM composition that reflects the needs of the cell.

The Golgi apparatus is the central organelle of the secretory pathway that receives cargoes (lipids and proteins) synthesized in the endoplasmic reticulum (ER), processes them (by adding sugars, lipids, or by proteases), and then sorts them to the correct intracellular compartment. The organelle consists of a stack of flat cisternae which vary from 3–12 in number depending on the cell type and organism [5]. In mammalian cells, each cell contains hundreds of such stacks that are interconnected to form the so-called Golgi ribbon that is localized near the centrosome. The Golgi stack is a polarized structure with the cargoes from the ER entering on one side of the stack and leaving from the other. The side of the stack that receives the cargo is called the cis-Golgi and the opposite side from where the cargo leaves the stack is called the trans-Golgi. The Golgi stack is sandwiched between networks of membrane on both the cis and trans-Golgi sides. These networks of membranes are appropriately called as cis-Golgi network (CGN) and trans-Golgi network (TGN) [6,7].

The cis and trans-poles of the Golgi, while structurally similar, show differences in terms of their lipid composition, the proteins localized to these structures, and also in the luminal environment of the cisternae [8,9]. Similarly, the enzymes that process the cargoes in the Golgi are not randomly distributed across the Golgi stack but are distributed in a manner that reflects their sequential mode of action on the cargoes that pass in the cis- to trans-direction [10]. A disturbance to this polarized localization affects the glycosylation process, especially those that involve competing pathways [11,12]. The luminal environment, especially pH, also changes as we go along the Golgi stack, from a pH of about 6.7 in the cis-Golgi to a more acidic pH of 6.0 in the trans-Golgi [9]. Finally, with regards to the lipid composition, there is a sharp transition in the type of lipids that predominate between the TGN and the rest of the Golgi stack or more specifically the cis-Golgi. The cis-Golgi and ER have a more phospholipid-rich environment, which facilitates the biogenic action of the compartments, while TGN reflects more the PM composition with enrichment of sphingolipids and cholesterol [8], which are essential for the barrier function of the PM.

In this review, we will discuss three distinct processes that happen at the Golgi that together build the cellular PM. These are: (1) Processing of protein and lipid cargoes in the Golgi; (2) Non-vesicular lipid transport through membrane contact sites (MCSs) at the TGN; and (3) Sorting of proteins and lipids at the TGN to the appropriate domain of the PM.

## 2. Cargo Processing at the Golgi Apparatus

The cargoes synthesized in the ER are transported to the Golgi apparatus where they are processed further by several post-translational modifications, of which the major ones are glycosylation and palmitoylation. There are also other cargo processing reactions in the Golgi, including phosphorylation and protease cleavage that are not discussed here. The processing reactions play an important role in determining the form, levels, and activity of the molecules in the PM.

Glycosylation is one of the most abundant post-translational modifications [13]. Glycans are polymers of monosaccharide units that can vary in the composition and sequence of monosaccharides they contain (10 monosaccharide residues are used for glycan biosynthesis in humans [14]), the anomeric linkage between the monosaccharides and further modifications, such as sulfation and acetylation of the sugars. The glycosylation machinery in the Golgi consists of glycosyltransferases (and glycosidases) that act sequentially to build a set of glycan polymers that are cell-type and cargo-specific [12]. Achieving this specificity in glycosylation, which is a template-independent process, depends on several factors of which the organization of the Golgi apparatus and the distribution of the enzymes across the Golgi cisterna play a very important role [12,15]. The sequential glycosylation reactions that build the glycans can be broadly divided into three steps–initiation, elongation and branching, and finally capping [14]. Based on the initiating glycosylation reaction, there are 16 distinct glycosylation pathways recognized to date in humans, including two lipid glycosylation pathways and 14 different protein glycosylation pathways [14]. We will discuss four of these major glycosylation pathways, including three protein glycosylation (N-glycosylation, mucin-type O-glycosylation, and glycosaminoglycan biosynthesis) and one lipid glycosylation (glycosphingolipid biosynthesis) pathway (Figure 1). The reader is referred to elsewhere for more in-depth reviews on the other significant glycosylation pathways [16,17]. While the initiating reactions are catalyzed by specific enzymes for each of the 16 distinct pathways, there is substantial overlap when it comes to the enzymes involved in the elongation of glycan chains and capping. The elongation reaction involves the action of a series of galactosyltransferases, N-acetylglucosaminyl (GlcNAc) transferases, and N-acetylgalactosamine (GalNAc) transferases. The capping or ending of the glycan chain is usually mediated by sialyltransferases and fucosyltransferases [14].

### 2.1. N-Glycosylation

N-glycosylation starts in the ER with the addition of a 14-sugar mannose-rich glycan onto the asparagine residue present within the consensus sequence Asn-X-S/T [18]. This mannose-rich glycan interacts with lectin-based chaperones in the ER and plays an important role in the folding of glycosylated cargoes [19]. The folded glycosylated cargo then is transported to the Golgi apparatus where it is first processed by mannosidases followed by a series of glycosyltransferases including GlcNAc transferases, galactosyltransferases, and sialyltransferases, in that order. The N-glycan can also be acted upon by fucosyltransferases [18]. Several of these enzymes compete with each other, especially at the level of addition of GlcNAc, so the resulting glycoform can vary depending on several factors, including the expression levels of enzymes, the milieu of the Golgi lumen, and the localization of the competing enzymes in the stack [12]. The early acting mannosidases and GlcNAc transferases are usually found in the cis-Golgi where the cargo enters the stack, while later acting enzymes, such as galactosyltransferases and sialyltransferases, are found in the trans-Golgi, where the final processing and exit of the cargoes take place. This ordered localization of the enzymes has been proposed to be important for the efficiency of the glycosylation reaction [10,15].

### 2.2. Mucin Type O-Glycosylation 

The typical O-glycosylation, the mucin-type O-glycosylation, starts in the Golgi with the addition of GalNAc to the O-atom of a Ser/Thr or rarely Tyr residue [20]. The consensus sequence that marks the O-glycosylation site is not clear. There are 20 GalNAc transferases (GALNTs) encoded in our genome that can initiate O-glycosylation [21]. They are distributed all across the Golgi stack and thus an O-glycan series can be initiated at any level of the stack [22]. Depending on the type of glycan added to the initiating GalNAc, there are four major types of O-glycan core structures that can then be further elongated [20]. The factors that determine the type of O-glycan produced still remain unclear. While the elongation is usually restricted to a few residues, it can go up to 20 residues and in the case of proteins such as mucins, the O-glycans can be densely packed to produce mucus-like secretions [20].

### 2.3. Glycosaminoglycans (GAGs)

GAGs are long polymers of repeating disaccharide units, one of which is an amino sugar (GalNAc or GlcNAc) and the other a galactose or uronic acid [23]. GAG chains can be very long and as a result, the proteins to which they are added are usually referred to as proteoglycans. The GAG production starts with the addition of xylose to a serine residue on the protein. The serine invariably has a glycine on the C-terminal side and two acidic amino acids are present either preceding the serine or following it, but a consensus sequence of initiation of the GAG chain is not known [23]. Xylose is then extended sequentially with two galactose and one glucuronic acid residue to result in the four sugar chain that is common to all the GAG chains. Depending on the sugars added further, there are at least two types of GAG chains–chondroitin sulfate/dermatan sulfate and heparan sulfate [23]. Chondroitin sulfate chains are polymers that have alternating glucuronic acid (GlcA) and GalNAc residues. In some cases, the GlcA can be epimerized to L-iduronic acid, and then the resulting GAG is called dermatan sulfate [23]. Heparan sulfate GAGs contain alternating GlcNAc and GlcA residues [23]. There is a distinct type of GAG called keratan sulfate or poly-N-acetyllactosamine that are polymers of alternating Galactose and GlcNAc residues. Unlike other GAGs, keratan sulfate polymers are usually added to N-glycans, O-glycans, and glycosphingolipids [23]. The GAG chains are usually sulfated at the TGN [24]. The sulfate on a GAG chain usually binds to positively charged amino acids on a protein; however, it is still unclear how the specificity of binding is achieved [25].

### 2.4. Glycosphingolipids (GSLs)

GSLs are key signaling molecules in the PM that regulate several physiological processes [26]. GSL biosynthesis starts in the ER with the production of ceramide, most of which is then transported by the ceramide transfer protein (CERT) to the TGN where it is converted to sphingomyelin [27]. The rest of the ceramide is either converted to galactosylceramide in the ER or is transported to Golgi by vesicular transport to be converted to glucosylceramide [26]. The glucosylceramide can be converted to several species of GSLs, including globosides, gangliosides, or lacto/neo-lacto GSLs [26].

The built glycans can also be further modified by phosphorylation, acetylation, sulfation, etc. creating several different glycoforms of a protein or lipid. The variations that are possible with glycans makes them information-rich and it is not surprising that they are used to communicate distinct cell states to the exterior, and as such, can act as cellular identity cards [27].

### 2.5. S-Palmitoylation

The Golgi apparatus is also the major location of another post-translational modification called S-Palmitoylation (Figure 1), which involves the addition of a fatty acid chain, mostly palmitate, to cytoplasmic cysteine residues through a thioester bond [28]. It is a highly conserved process occurring in all eukaryotes and is mediated by a family of palmitoyltransferases. Palmitoyltransferases are multi-pass transmembrane proteins (usually have four transmembrane domains) with a conserved DHHC tetrapeptide (Asp-His-His-Cys), critical for their enzymatic activity, in a cysteine-rich domain present on the cytosolic side [29]. In mammals, S-palmitoylation is catalyzed by a family of 23 enzymes (hereafter DHHCs) most of them localized to ER and/or Golgi [30]. Among the 19 DHHCs that localize to ER and/or Golgi, nine are Golgi specific (DHHC3,7,9,11,13,15,17,21,22) [30,31,32]. Interestingly, six out of nine Golgi-specific DHHCs co-localize with cis-Golgi markers, whereas the remaining three were detected at the trans-Golgi [32]. Among all the lipid modifications, S-palmitoylation is the only one that is reversible [33]. The palmitate removal from proteins in the cytosol is mediated by acylprotein thioesterases (APTs), which are ubiquitous cytosolic thioesterases responsible for most of the depalmitoylation processes. The human genome encodes for two APTs-APT1 and APT2 [34]. Moreover, several a/b-hydrolase domain (ABHD) proteins have been proposed to catalyze depalmitoylation reactions [35]. As a reversible lipid modification, S-palmitoylation regulates reversible and dynamic attachment of proteins to membranes and so plays an important role in regulating signaling pathways.

Thus, the Golgi apparatus receives cargoes synthesized in the ER and processes them to produce a variety of modified gene products that are not directly encoded in the genome.

## 3. Non-Vesicular Lipid Transport through MCSs at the TGN

A distinguishing feature of membranous compartments is their differential lipid composition in addition to their characteristic protein profile. For instance, the PM is highly enriched in cholesterol and sphingolipids while the ER where the biosynthesis of the lipids happens or at least begins, is depleted in these lipids [36]. There is also a close match between the lipid composition of a compartment and the transmembrane domains of the proteins localized to the compartment [36]. While the distinctive localization of proteins to various compartments is achieved by the vesicular transport machinery (see below), the lipids do not solely depend on this machinery for their differential localization. Lipid transport proteins that bind and transfer individual lipids across MCS play an important role in this process [37]. MCS is defined as close apposition between the membranes of two organelles (<30 nm apart) that promotes the tethering of the membranes but not fusion so that the individual identity of the organelle is maintained [38]. The major membrane contact site operating at the secretory pathway is the one between ER and the TGN. This ER-TGN contact site plays an important role in transferring lipids that are essential for building the PM [39].

A recent study on the ER-TGN contact sites has shown that nearly half of the Golgi stacks in a mammalian cell have these MCSs, and about a quarter of the TGN surface area is engaged in contact site formation [40]. Thus, we can conclude that extensive areas of TGN are involved in contact site formation. The ER-TGN contact sites are maintained by tethering action of the ER-localized VAMP-associated proteins or VAPs (VAP-A and VAP-B) [40]. VAP proteins play an important role in several other MCSs too [41]. VAP proteins bind to several TGN localized lipid transfer proteins (LTPs) to promote tethering of the ER to TGN. These LTPs usually have a pleckstrin homology (PH) domain that binds to phosphatidylinositol-4-phosphate on the TGN and a FFAT (two phenylalanines in an acidic tract) motif that binds to VAP proteins, thus bridging the ER to TGN [38]. In addition to acting as a tether, the LTPs also have a domain that specifically recognizes lipids that they transfer. For instance, ceramide transfer protein (CERT) has a steroidogenic acute regulatory protein (StAR)-related lipid transfer (START) domain that binds and transfers ceramide from ER to TGN [38]. Similarly, Oxysterol binding protein 1 (OSBP1) and Oxysterol binding protein-related protein 9 (ORP9) have OSBP related domains (ORDs) that are involved in cholesterol transfer, while ORP10 and OPR11 have ORDs involved in the transfer of phosphatidylserine (Figure 2a) [38]. Transfer of these lipids from ER to TGN against a concentration gradient (the lipids are usually concentrated in TGN) is achieved by two mechanisms: immediate modification of the lipid at the TGN to eliminate the concentration gradient as in the case of ceramide transfer, where ceramide is efficiently converted to sphingomyelin by TGN localized sphingomyelin synthase 1 (SMS1) [27]; and coupling the transfer of these lipids from ER to TGN to the countercurrent transfer of PI4P from TGN to ER which is also concentrated in the TGN. This happens in the case of the transfer of cholesterol and phosphatidylserine from ER to TGN [42]. The PI4P transferred to ER is efficiently hydrolyzed to phosphatidylinositol by ER-localized SAC1 phosphatase [42]. This helps maintain the PI4P gradient across the TGN-ER junction to in turn drive the transport of other lipids from ER to TGN. The lipids that are thus concentrated are then assembled along with the PM targeted proteins in the TGN before being sorted to the appropriate compartment by the TGN sorting machinery.

## 4. Sorting of Cargoes at the TGN

The cargoes that have been processed in the Golgi apparatus reach the TGN where some capping reactions that terminate the glycan processing happen before the sorting of cargoes to appropriate compartments [43,44]. While the major segregation of cargoes happens in the TGN, some cargoes are segregated already at the level of the Golgi stack, especially the lysosomally targeted cargoes [45]. But whether this allows for their differential processing is not known. There are several known pathways that transport cargoes out of the TGN. These include sorting of cargoes to endosomes/lysosomes, to secretory granules, to apical or basolateral PM in polarized epithelial cells (Figure 3), and to axons or dendrites in neurons [46]. The sorting of cargoes to these distinct compartments at the TGN is based on signals the cargoes expose on their cytosolic side [43,46]. Luminal proteins bind to transmembrane cargo receptors that have the signals for sorting machinery. The best understood of these cargo receptors is the mannose-6-phosphate receptors (M6PRs) that bind the lysosome targeted proteins that have the mannose-6-phosphate signal [47]. There are two M6PRs in the cell—cation-dependent M6PR and cation-independent M6PR that sort a non-overlapping set of lysosomal cargoes at the TGN [47]. There are also other receptors operating at the TGN including Sortilin (which sorts lysosomal enzymes Cathepsin D and H), Sortilin related receptor with A-type repeats (receptor highly expressed in neuronal cells and involved in the sorting of amyloid precursor protein), lysosomal integral membrane protein 2 (LIMP2 involved in the sorting of β-glucocerebrosidase) and others [43]. The sorting signals on the cargo receptors or the transmembrane cargoes are recognized by adaptor proteins that couple these proteins to coat proteins, usually clathrin, to form the carriers. There are several adaptor proteins acting at the TGN, including the heterotetrameric AP (adaptor protein) family members and the monomeric Golgi-localized, γ-ear-containing, Arf (ADP-ribosylation factor)-binding (GGA) proteins [43,48]. There are five known AP protein complexes of which 3 act at the TGN—AP-1, AP-3, and AP-4 [43]. AP-1 and AP-3 complexes bind clathrin coats, AP-4 does not [43]. The AP complexes consist of 4 subunits—two large subunits (A, B), one medium subunit (M), and a small subunit (S). Depending on the isoform of M subunit incorporated AP-1 exists as AP-1A and AP-1B complexes [43,48], and depending on isoforms of B and M subunits incorporated, AP-3 exists as AP-3A and AP-3B complexes [43]. While μ1A subunit is ubiquitously expressed, μ1B is expressed predominantly in epithelial cells [43,48]. The AP complexes bind to tyrosine and leucine-based signals present on the cargo or cargo receptor molecules. The tyrosine-based motif consists of a consensus sequence YXXϕ, where Y represents tyrosine, X any amino acids, and ϕ a hydrophobic amino acid [43]. Similarly using the single letter amino acid code, the consensus sequence of the leucine based motif is [DE]XXXL[LI] [43]. While all AP complexes recognize these signals, their preferential binding to certain cargoes is influenced by the nature of the X residue, the ϕ residue, and the amino acid residues surrounding the consensus motif. For example, AP-3 preferentially binds cargoes containing YXXϕ motif where the consensus motif is preceded by glycine and followed by acidic amino acids [43]. Similar differential interaction of AP complexes with dileucine residues has also been observed [43]. The AP-1 complex usually transports cargoes to the basolateral PM while AP-3 to lysosomes, since the lysosomal cargoes tend to have YXXϕ that preferentially binds AP-3 [43]. The monomeric adaptor GGA has three isoforms in humans—GGA1, GGA2, and GGA3 [49]. The GGAs have a N-terminal VHS (Vps27, Hrs, STAM) domain that binds to dileucine motifs with acidic amino acids on the N-terminus [49]. The DXXLL consensus sequence that binds GGA is usually found in cargoes destined to the endosomes [43]. GGAs have also been shown to bind ubiquitin to sort ubiquitinated cargoes to the endosomes [43].

The promotion of the interaction between adaptors and cargoes is regulated by Arf GTPases and PI4P present on the TGN membrane. The recruitment of adaptor complexes by Arf GTPase can also lead to their activation by a conformational change that promotes cargo binding as in the case of AP-1 [43]. Once recruited, the adaptor molecules bind to cargoes and sort them into specific exit sites, where they also promote the recruitment of clathrin coats for cargo sorting and carrier formation [48]. Finally, PI4P-Arf machinery also recruits the fission machinery to the budding tubules to promote carrier fission [50,51].

While this represents the classical understanding of how cargo is sorted at the TGN, there are also alternate ways to form cargo laden carriers at the TGN. A well-known example is the calcium-mediated cargo sorting at the TGN. The actin-cofilin machinery regulates the activity of the calcium pump SPCA to pump calcium into the lumen of the TGN, which then promotes the binding of cargoes to CAB45 and their aggregation [52]. The aggregated cargoes then associate with sphingomyelin on the TGN membrane to promote carrier formation [52]. There are also other similar but distinct pathways operating at the TGN to transport cargoes. These include sphingomyelin-based sorting of lipoprotein lipase [52], calcium based sorting of secretory granules [52], a PKD and Arfaptin dependent formation of chromogranin A carriers [53], and PKD and microtubule-dependent formation of carriers termed as “carriers of TGN to cell surface” (CARTS) [54,55,56]. The complexity of pathways operating at the TGN requires control systems that can regulate these pathways to favor optimal sorting and transport. Recently, one such control system consisting of an autoregulatory signaling pathway that responds to the presence of cargo and activates PKD at the TGN has been identified. This control system regulates the PKD dependent exit of basolateral cargoes from the TGN [57].

While distinct signals that transport cargoes to basolateral PM have been identified, the signals for the transport to apical membrane are still unclear. An exception is the apical sorting signal on Rhodopsin A, the VXPX motif that binds Arf4, which then organizes a complex of Rab11, FIP3, and ASAP1 to promote the transport of rhodopsin from TGN to the ciliary membrane present on the apical PM [43]. While a clear signal has yet to be identified on most other apical cargoes, a recurring determinant for apical transport appears to be polymerization or concentration of cargoes. This is achieved by cargo affinity for the lipid rafts, concentration by Galectins that bind to cargo glycan modifications [43], or by calcium-mediated aggregation [58]. However, it is still unclear how this polymerization leads to carrier formation and what molecules are involved.

## 5. Golgi Functions Regulate PM Composition and Activity

### 5.1. Golgi Processing

The glycosylation reactions happening at the Golgi determine the glycoform of a protein that is present at the PM. A change in the glycoform influences several aspects of a protein function, including its interactions with the other proteins/molecules, its association with lipid rafts, its ability to polymerize, and its half-life [58]. A pertinent example of how glycans influence protein function by controlling its PM levels comes from the study of N-glycan processing in the Golgi apparatus. N-glycans are processed in the Golgi apparatus initially by glycosidases followed by GlcNAc transferases or monoacylglycerol acyltransferases (MGATs). There are several MGATs acting on N-glycans in the Golgi. MGAT1 and MGAT2 process the N-glycan by adding GlcNAc residue in a β-1,2 linkage that results in biantennary glycan chains; MGAT5 adds GlcNAc residue to mannose in the β-1,6 position that leads to tri and tetraantennary N-glycans [18]. MGAT1 has a higher affinity for UDP-GlcNAc (activated form of the sugar that is used as a donor in glycosylation reactions) than MGAT5 and so can glycosylate even at lower luminal concentrations of the sugar donor. As the concentration of UDP-GlcNAc increases in the Golgi lumen due to increased cellular hexosamine levels, MGAT5 becomes active and adds β-1,6 linked branches to N-glycans [59]. This increased branching of N-glycan promotes interaction of the protein with galectins present in the extracellular milieu and prevents their endocytosis thus increasing their half-life on the PM [59]. Surprisingly, the growth factor receptors tend to have more N-glycan sites than the growth antagonizing receptors and so increased branching of receptor associated N-glycans differentially promotes the PM retention of growth-promoting receptors compared to growth-inhibiting receptors and thus promotes cell growth [59]. This regulation provides an excellent example of how Golgi processing can affect the glycoform of a protein present on the cell surface and consequently cellular signaling and physiology.

Changes in the glycoform can also affect protein–protein interactions. For instance, the epidermal growth factor receptor (EGFR) is known to have 12 N-glycans on the extracellular domain that are processed as the receptor passes through the Golgi. Sialylation and/or fucosylation of the N-glycan branches prevents EGFR dimerization and subsequently reduces receptor kinase activity (Figure 4) [60]. On the contrary, core fucosylation of the EGFR N-glycans promotes the interaction of EGFR with its ligand and promotes receptor activation [60].

Receptor activities can also be influenced by changes in the glycosylation of PM-associated lipids. The amount of GSLs and their glycosylation is known to affect the activities of PM receptors by direct interaction of lipids with receptors or indirectly by sorting them into or away from lipid nanodomains/rafts. An example of the former is the regulation of EGFR activity by the ganglioside GM3 (and similar sialylated gangliosides such as GM1). The interaction of the sialyl group of GM3 with a lysine at a membrane proximal position on EGFR prevents receptor dimerization, while an interaction between the sialyl group of GM3 with the GlcNAc on the N-glycans of the EGFR prevents its autophosphorylation and activation [61]. GSLs act on several growth factor receptors in a similar manner and the GM3S KO mice have enhanced insulin sensitivity due to enhanced phosphorylation of insulin receptor [62]. In addition to alteration of signaling by direct binding to receptors, GSLs can also alter the lipid raft association of receptors likely through their hydrophobic ceramide chains. Indeed, during contact-mediated inhibition of cell proliferation, there is an increased GM1 biosynthesis and this increased GM1 is known to alter the distribution of EGFR in lipid rafts, which in turn, alters the receptor-mediated signaling [63].

Similar to glycosylation, palmitoylation can also influence the localization and consequently the function of a protein. Analysis of palmitoyl-proteomes reveals that almost 20% of the human proteome including both transmembrane and peripheral membrane proteins is S-palmitoylated [64]. S-palmitoylation is the most frequently observed lipid modification in neurons and plays a crucial role in synaptic function [33,65]. S-palmitoylation has been shown to influence protein folding, degradation, formation of protein complexes, protein sorting, and association with lipid rafts [66]. Protein sorting following S-palmitoylation is a complex phenomenon, with palmitoylation having different effects on different proteins. For instance in neurons, S-palmitoylation of AMPA receptors and glutamate-gated ion channels GluA1 and GluA2 by DHHC3 causes their accumulation in the Golgi and a reduction of AMPA receptors expression at cell surface (Figure 4) [67,68]. On the contrary, lipoprotein related protein 6 (LRP6) a crucial component of the Wnt signaling pathway requires palmitoylation to be exported out of the secretory pathway onto the PM [69]. In addition to the localization to specific membrane compartments, palmitoylation also plays a role in the sorting of transmembrane proteins into lipid rafts, and here again, opposing effects of palmitoylation have been reported. For instance, DHHC3 dependent palmitoylation of death receptor 4 is required for its localization to lipid rafts and signaling downstream for ligand-induced cell death [70]. On the other hand, palmitoylation negatively affects the lipid raft localization of anthrax toxin receptor tumour endothelial marker 8 [71]. Given these opposing effects, the mechanistic basis of palmitoylation-based sorting is likely to be protein specific.

Cycles of palmitoylation-depalmitoylation are also crucial in controlling the localization and function of various peripheral membrane proteins [72]. For instance, N/H-Ras localization and signaling from PM and Golgi critically depends on the dynamics of palmitate turnover [73,74]. These two small GTPases are palmitoylated by DHHC9 in the Golgi at either one (N-Ras) or two (H-Ras) cysteine residues. This lipidation mediates their localization to PM where APT1- and APT2-mediated depalmitoylation determines their return to the Golgi, allowing another round of palmitoylation [74,75,76]. Growth factor stimulation normally leads to rapid activation of H-Ras at the PM and delayed but sustained activity on the Golgi [73,77]. Therefore, the growth factor–induced Ras signaling response requires localization of Ras at both PM and Golgi. Blocking the palmitoylation cycle by the APT1 inhibitor Palmostatin B does not reduce H-Ras activation at the PM, but reduces the retrograde transport of H-RasGTP to the Golgi, altering the connection between distinct sets of downstream signaling cascades that make the Ras-signaling response. Thus, the processing of PM components as they pass through the Golgi apparatus determines their localization to the PM, their activity and finally signaling from the PM.

### 5.2. Non-Vesicular Lipid Transport

Transport of lipids against a concentration gradient at the MCS contributes significantly to the concentration of cholesterol and sphingolipids at the TGN, which is essential for the assembly of carriers with a lipid composition mimicking that of the PM. The importance of these processes in determining the PM lipid composition was shown by cell lines mutated for CERT. As mentioned earlier, CERT transports ceramide in a non-vesicular fashion from ER to the TGN where it is converted to sphingomyelin by the action of SMS1 [27]. Impairment in CERT activity by mutation reduces SM production as well as SM levels in the PM as measured using a specific probe lysenin [27]. Lysenin is a bacterial toxin that binds to SM on the PM to induce cell death and the reduction in SM levels in the PM following impaired CERT activity makes the mutant cells less sensitive to lysenin action [78]. Similarly, another lipid transfer protein FAPP2, which mediates the transport of glucosylceramide when impaired, reduces GSL biosynthesis and this is reflected as reduced levels on the PM (as measured again by specific bacterial toxins that bind to them) [79]. Reducing OSBP1 levels is known to impair lipid metabolism but whether it also leads to a reduction in the surface levels of cholesterol is not clear. However, it does change the composition of PM by affecting the exit of cargoes from the Golgi apparatus. Reduction in OSBP1 has been shown to block the exit of Caveolin-1 from the Golgi [80]. Reduction in OSBP1 was also shown to mislocalize the SNARE proteins GS28 and GS15 which are involved in transport across the Golgi apparatus and also reduce the levels of mannosidase II and thus likely affecting N-glycan processing in the Golgi [80]. On the contrary, increasing OSBP1 levels downregulated the amyloidogenic processing of Amyloid precursor protein (APP), while decreasing OSBP1 produced the opposite effect [81]. A role for ER-TGN MCS in cargo transport out of the Golgi was shown decreasing the levels of FAPP1 which increases TGN PI4P levels and consequently promotes secretion of selected cargoes like ApoB100 [82]. The role of MCS in regulating the transport of cargoes from TGN was also demonstrated by the study which showed that cholesterol regulator SCAP controls the biogenesis of CARTS carriers from the TGN by disorganizing the ER-TGN MCS [83]. Thus, the activities of the ER-TGN contact sites determine PM composition by regulating the transport and production of PM lipids and also by regulating cargo exit from the TGN.

### 5.3. Sorting at TGN

After the cargoes are processed at the Golgi apparatus, they are sorted to their correct destination at the TGN. This sorting step is a key player in determining the differential PM composition in polarized cells like epithelial and neuronal cells. Mutation of targeting signals present in the cargoes targeted to the basolateral PM has been shown to result either in their targeting to apical PM or a loss in their differential localization to the basolateral PM [84]. The expression of key players of the TGN sorting machinery also determines the differential composition of PM by regulating the targeting of molecules. For instance, MDCK and LLC-PK1 epithelial cell lines, derived from kidney distal and proximal tubule cells, respectively have differential expression of μ1B subunit. MDCK is μ1B positive but the expression of the subunit is absent in LLC-PK1 cells [48]. While both the cells can form a polarized monolayer in vitro they differ in the localization of some proteins. TfR and LDLR are basolaterally localized in MDCK cells but show an apical localization in LLC-PK1 cells. This difference is mainly due to difference in the expression pattern of μ1B since expressing μ1B cells in LLC-PK1 leads to the basolateral localization of the two receptors [48].

The importance of the TGN sorting machinery in determining the PM composition is also evident from mice KO experiments. Mice KO for μ1B show impaired basolateral sorting of several cargoes like LDLR, Ephrin receptor, E-Cadherin, and IL-6 signal transducer [48]. Missorting of E-Cadherin leads to activation of β-catenin dependent pathway that results in hyperplasia of the intestinal epithelial layer and development of duodenal polyps in this mouse [85]. The missorting of IL-6 signal transducer impaired immune signaling pathways resulting in increased penetration of commensal bacteria into the intestinal mucosa and inflammatory colitis [86]. Similar missorting of proteins and impaired epithelial function upon inactivation of AP-1 subunits were also observed in *C. elegans* and zebrafish [43].

In addition to epithelial cells, neuronal cells also have polarized PM domains with the basolateral PM of an epithelial cell most likely corresponding to the somatodendritic PM and the apical PM to the axonal PM. Impairment of AP-1 subunits or clathrin led to impaired targeting of somatodendritic proteins like transferrin receptor and NMDA-type glutamate receptor protein [48]. Impairing AP-4 activity led to mislocalization of dendritic PM proteins AMPA-type glutamate receptors, Delta2 glutamate receptor and LDLR [48]. TGN sorting also plays an important role in the formation of Planar cell polarity, which involves the asymmetric distribution of proteins within the plane of the epithelium with proteins like Vangl2 and Frizzled6 distributed to opposing cellular boundaries. Vangl2 and Frizzled 6 are sorted into distinct carriers at the TGN, with the sorting of VAngl2 dependent on Arfrp1/Arl3 and AP-1, while Frizzled 6 using another unidentified machinery [43].

Thus, processing and sorting of proteins and lipids at the Golgi apparatus can influence their localization to the correct PM.

## 6. Diseases Resulting from Impaired Golgi Function

### 6.1. Golgi Processing

Changes in Golgi processing reactions result in inappropriate forms of proteins and lipids being displayed on the PM, which influences several aspects of the interaction of a cell with its environment. These changes when detrimental to development and/or physiology result in diseases, including genetic diseases and cancer. A group of genetic diseases resulting from impaired Golgi glycosylation reactions resulting in developmental problems is known as congenital disorders of glycosylation (CDG) [87,88]. There are nearly 140 CDGs described to date and most cases involve an autosomal recessive mutation in glycosylation enzymes or trafficking regulators of the Golgi [87,88]. These diseases are usually associated with intellectual disability, impaired development, and also impaired liver functions suggesting a major role for glycosylation in the functioning of these organs [87,88]. While these diseases are associated with impaired glycosylation in the Golgi apparatus, the molecular details of how a change in glycosylation leads to the pathology is not well understood. On the other hand, the contribution of changes in glycosylation to cancer has been studied in relatively more detail. Changes in glycosylation associated with cancers have been noted decades ago and, in several cases, the biomarkers used for cancer diagnosis are glycans or glycoproteins [89]. While the molecular mechanism of how this change is brought about is not clear, at least in some cases the mechanisms of the change and the molecular details of the consequences have been dissected. It was noted decades ago that oncogenic transformation is associated with an increase in cell surface sialylation. This sialylation, which increases the negative charge, was proposed to reduce cell-cell interaction due to negative charge induced repulsion and contribute to carcinogenesis by disorganizing cell monolayers and by also affecting the recognition of cancer cells by the immune system [90,91]. Recently, the activity of ST6GAL1 was found to be increased in cancer tissues and the increased ST6GAL1 activity was shown to be associated with several hallmark properties of cancer [92].

A well-known biomarker of cancer is the increase in Tn antigens on the cancer cell surface [89]. Tn antigen corresponds to the O-linked GalNAc groups on proteins that are added by the O-glycan initiating enzymes—GalNAc transferases or GALNTs. It was shown that hypermethylation leads to decreased expression of a chaperone that folds enzymes involved in O-glycan elongation. The absence of these elongation enzymes results in increased Tn antigen expression in cancers, which were shown to directly induce oncogenic features including cell growth and invasion [93,94]. It was also shown that activation of the oncogene Src leads to increased retrograde trafficking from Golgi to ER of GALNTs, which results in increased O-glycosylation of proteins in the ER and increased surface levels of Tn antigen [95]. The increased O-glycosylation is known to affect the localization and activity of proteins, such as calnexin and matrix metalloproteinases (MMPs) that promote metastasis [96,97].

While O-glycosylation of MMPs shows how glycosylation of a protein can affect its activity directly, the dissection of the pathogenic pathway of the oncogene *GOLPH3* has shown how glycosylation can indirectly affect the activity of PM receptors. *GOLPH3* was recognized to be an oncogene but its mode of action was not clear [98,99]. Recently it was shown that GOLPH3 prevents the exit from the Golgi of a key enzyme of the GSL biosynthetic pathway lactosylceramide synthase (LCS) and thus prevents its degradation [100]. Increased levels of GOLPH3 following copy number alteration results in increased levels of LCS in cancer cells, which in turn leads to increased GSL biosynthesis [100]. It was observed that these increased GSL levels led to increased activation of growth factor receptors on the PM (Figure 5) [100]. Whether it is due to the direct interaction of GSLs with the receptors and indirectly through their sorting into lipid raft-like nanodomains is not clear. Nevertheless, these examples indicate how alteration of Golgi glycosylation reactions change the glycoforms of PM constituents resulting in pathological signaling from the PM.

Similar to glycosylation, aberrant palmitoylation is also associated with several pathologies, including neurological diseases and cancer [101]. For instance, loss-of-function mutations in DHHC9 and altered palmitoylation of its substrates have been identified in patients with neurodegenerative diseases, neurodevelopmental disorders, and patients with X-linked intellectual disability (XLID) [65,102,103,104]. DHHC9 plays an important role in the balance between excitatory and inhibitory synapses by controlling the PM localization and/or stabilization of two small GTPases, N-Ras and TC10 [105]. DHHC9 palmitoylates N-Ras and targets it to the PM leading to the activation of extracellular signal-regulated kinase (ERK) signaling and suppression of c-Jun N-terminal kinase (JNK) signaling pathways [105]. Silencing of DHHC9 in hippocampal cultures results in reduction of N-Ras palmitoylation and ERK activation, and an increase of JNK activation, whereas DHHC9 overexpression produces an opposite output [105]. Similarly, DHHC9-mediated palmitoylation of TC10 is crucial for its localization to and stabilization on the PM where TC10 promotes formation of inhibitory synapses [105]. Therefore, DHHC9-mediated palmitoylation of Ras and TC10 controls the activation of distinct molecular pathways in neurons, which trigger two distinct processes, dendrite outgrowth (Ras pathway) and inhibitory synapses (TC10 pathway). Thus, DHHC9 loss-of-function mutations observed in XLID patients have been suggested to be causative for their epileptic comorbidities (Figure 5) [105]. Detrimental effects of aberrant palmitoylation have also been associated with the loss-of tumour suppressive activities of SCRIB observed in several cancers [106]. DHHC7 is the major palmitoyltransferase for SCRIB and palmitoylation is required for its cell-junction localization. On the PM, SCRIB activates Hippo pathway causing cytoplasmic sequestration of YAP and TAZ, major Hippo cascade effectors with oncogenic activities [107,108,109]. Knocking-down DHHC7 in MCF10A human breast epithelial cell line leads to mislocalization of SCRIB and an accumulation of YAP in the nucleus, where it promotes transcription of its target genes that contribute to oncogenesis. SCRIB is frequently amplified and mislocalized in tumours, such as ovarian and prostate cancers [106]. Considering that DHHC7 is frequently deleted in those and several other cancers, SCRIB palmitoylation and its PM localization could be crucial for SCRIB tumour-suppressive functions [106]. Thus, processing of proteins and lipids at the Golgi apparatus is essential to determine the proper form of the protein on the PM and alters the signaling from the PM with pathological consequences.

### 6.2. Non-Vesicular Lipid Transport

The activity of ER-TGN MCS has bearing on the physiology of the cell and the organism. Recently a mutation in CERT (S135P) was described that is associated with intellectual disability [110]. It was shown that this mutation increased the activity of CERT and thus leads to increased sphingomyelin biosynthesis (Figure 2b) [111]. How the altered SM levels leads to the phenotype is not clear, though it likely affects PM properties and activities. Similarly, CERT levels were found to be reduced in triple-negative breast cancers (TNBC) and this leads to reduced PM sphingomyelin levels, which alters the properties of the PM [112]. This leads to alterations in the mobility of EGFR/ErbB1 in the PM, its ligand induces autophosphorylation, internalization, and chemotaxis (Figure 2c) [112]. These studies suggest that the non-vesicular lipid transport at the MCS plays an important role in maintaining the PM lipid levels within a physiologically acceptable range and any disorganization in this regard results in a lipid imbalance that has pathological consequences.

### 6.3. TGN Sorting

Impaired TGN sorting leads to improper PM composition that in several instances results in pathological consequences. For example, the distal determinant for basolateral targeting of LDLR is mutated (G823D) in patients with familial hypercholesterolemia–Turku variant [84]. This mutation leads to missorting of LDLR to apical surface of epithelial cells and bile canicular surface in hepatocytes. The decreased presence of LDLR in basolateral surface results in reduced cholesterol clearance from blood leading to hypercholesterolemia (Figure 3) [84]. The mutation is μ1A and μ1B subunits of AP-1 lead to MEDNIK syndrome and fried-type X-linked mental retardation respectively (Figure 3) [48]. These diseases are associated with common symptoms, including intellectual disability and neurological abnormalities. In addition, likely due to the subunit specificity of signal binding, they also have some specific symptoms. MEDNIK, for instance, is also associated with enteropathy and skin disorders [48]. These consequences are likely due to the impaired polarized sorting of cargoes mediated by AP-1 subunits, but needs experimental evaluation. Mutations in the subunits of AP-4 cause neurodevelopmental disorders with characteristics of hereditary spastic paraplegia in humans [48]. This disease is associated with microcephaly, cerebral palsy, and profound intellectual disability. It has been shown that in the purkinje neurons of these patients, Delta2 transport to dendrites is decreased [48]. Studies have also shown that colonic mucosa from Crohn’s patients showed reduced levels of μ1B mRNA and some colorectal cancer tissues also have shown reduced μ1B mRNA [48]. While these changes are consistent with the observed pathology based on mouse KO studies, whether they contribute to the pathology needs to be investigated.

## 7. Conclusion and Future Directions

The discussions above point to the fact that the Golgi apparatus plays an important role in determining the composition and organization of the PM. Through this, the Golgi apparatus controls how the cell interacts with the external environment—by regulating the signals sent out by the cell and also the signals received by it from the outside. Though not discussed in this paper, the Golgi apparatus also regulates the composition of the extracellular matrix in multicellular organisms, which have important physiological and pathological consequences [113]. While decades of research has led to an understanding about the functioning of the Golgi apparatus and how it organizes cellular membranous compartments, several questions remain unanswered, which have implications for PM composition and activity:

1. How is the transport and processing of cargoes in the Golgi apparatus coordinated to achieve faithful processing of the cargoes?

2. What are the molecular mechanisms that organize the Golgi apparatus and determine the sub-Golgi localization of the glycosylation machinery?

3. How many transport pathways exit from the TGN?

4. How are the cargoes destined for apical transport recognized? What are the molecular machineries involved in apical transport?

5. How are all these processes regulated? What are the autoregulatory pathways maintaining homeostasis of the compartments? How does the Golgi apparatus respond to external signals to coordinate its functions efficiently?

6. There are still numerous proteins localized to the Golgi apparatus whose function is not completely understood, such as several TMEM proteins and Golgins. How do they contribute to the functioning of the Golgi?

7. While we do have a model for how each member of the trafficking machinery contribute to transport across the organelle, the complications arising from studies of CDG—different mutations in the same gene causing different phenotypes, mutations in genes encoding proteins that belong to the same complex having varied phenotypes—have shown that our understanding of the system is far from complete and we still need to build a model of Golgi transport that explains all these divergent phenotypes.

In the coming years, a combination of systems level analysis of Golgi functions along with the classic methods of focused and reductive analysis of individual components will likely lead to better understanding of Golgi functions and its contribution to physiology and pathology.

## Figures and Tables

**Figure 1 cells-11-00368-f001:**
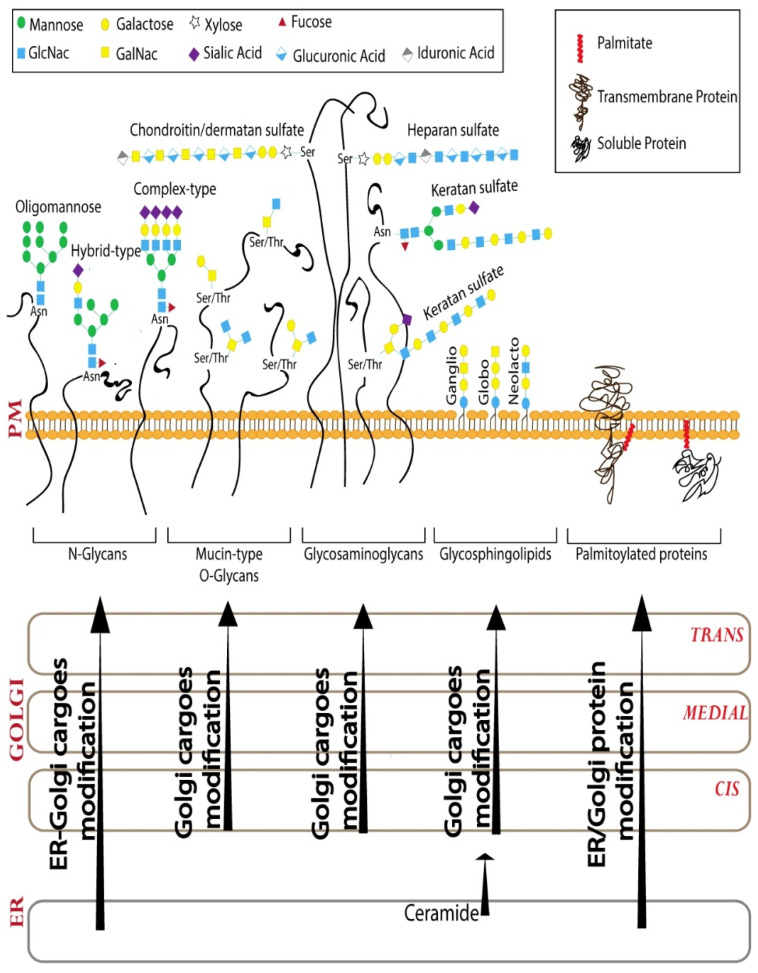
Cargo processing in the Golgi. N-glycosylation, mucin-type O-glycosylation, glycosaminoglycan, and glycosphingolipid pathways are 4 of the 16 distinct glycosylation pathways that modify cargoes passing through Golgi and modulate signaling from PM. Golgi apparatus is also the major palmitoylation location in the cell. While some of the processing reactions happen exclusively in the Golgi apparatus, some of them are shared between ER and Golgi as shown.

**Figure 2 cells-11-00368-f002:**
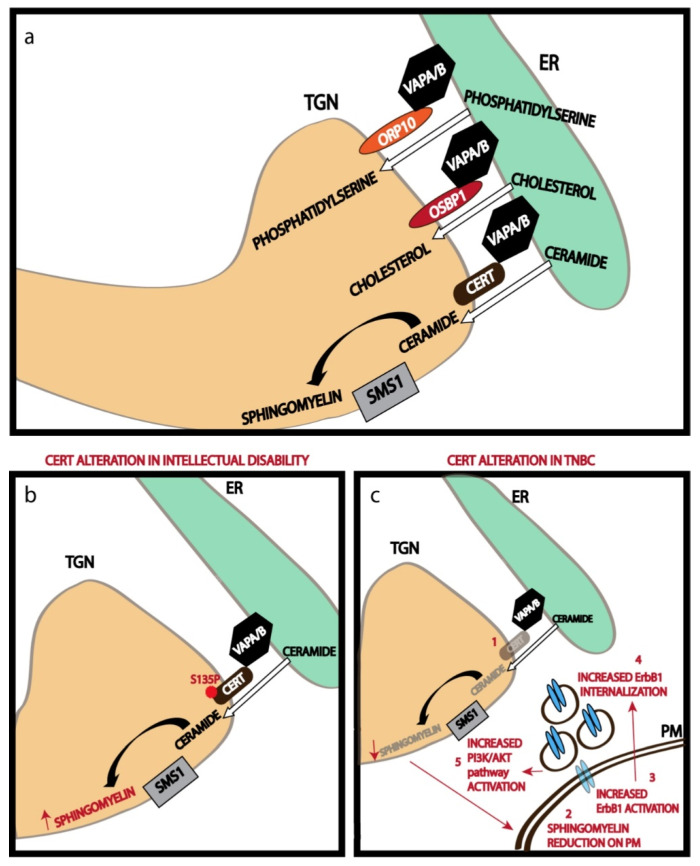
Non-vesicular transport of lipids at the TGN. (**a**) LTPs acting at the ER-TGN MCSs transfer the indicated lipids from the ER to TGN, where they can be further metabolized as in the case of ceramide which is converted to Sphingomyelin. (**b**) A mutation in CERT hyperactivates it, thus increasing the production of Sphingomyelin, which is associated with intellectual disability. (**c**) CERT alteration in triple negative breast cancer cells reduces its activity and hence Sphingomyelin production, which leads to a cascade of events culminating in increased PI3K/Akt activation.

**Figure 3 cells-11-00368-f003:**
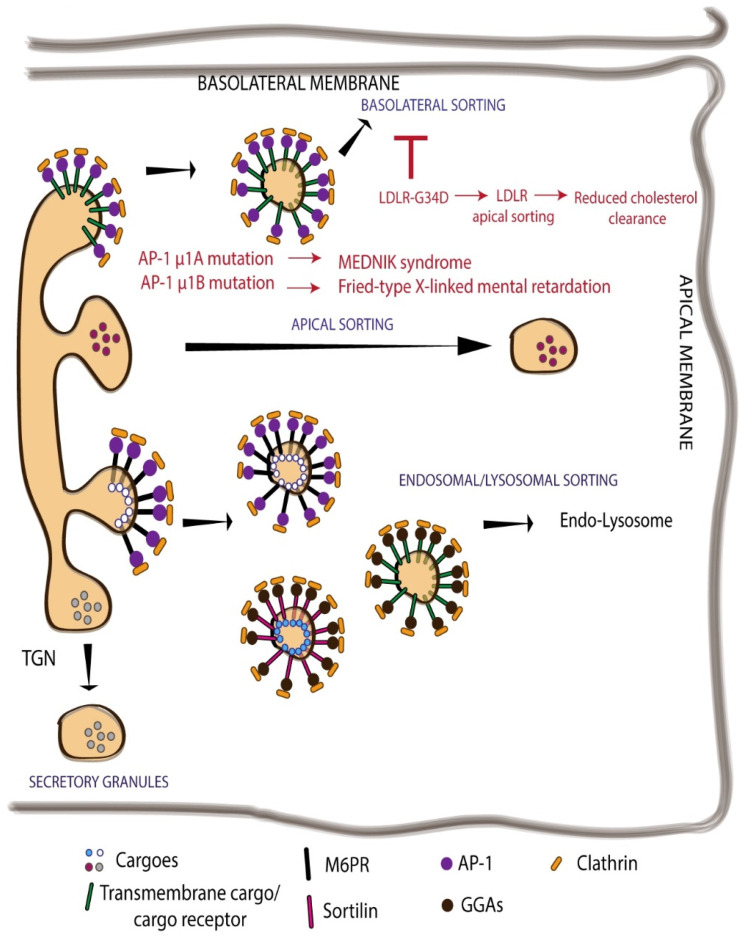
TGN sorting affects PM composition. Cargoes modified in the Golgi are correctly sorted to their final destination at the TGN. Several TGN exit routes have been reported. Two well-described sorting mechanisms at TGN are the receptor-mediated sorting of lysosomal hydrolases via M6P receptor (M6PR) to the lysosome, and the sorting of soluble proteins via sortilin to endosomes, both in clathrin-coated vesicles. Transmembrane proteins or constitutively secreted cargo can be transported towards the apical or basolateral PM. In professional secretory cells, proteins leave the TGN in secretory granules that fuse with the PM and release their content into the extracellular space. Mutations of key players that mediate protein sorting at the TGN, such as adaptor proteins or targeting signals in the cargoes, have been associated with pathological conditions (in red).

**Figure 4 cells-11-00368-f004:**
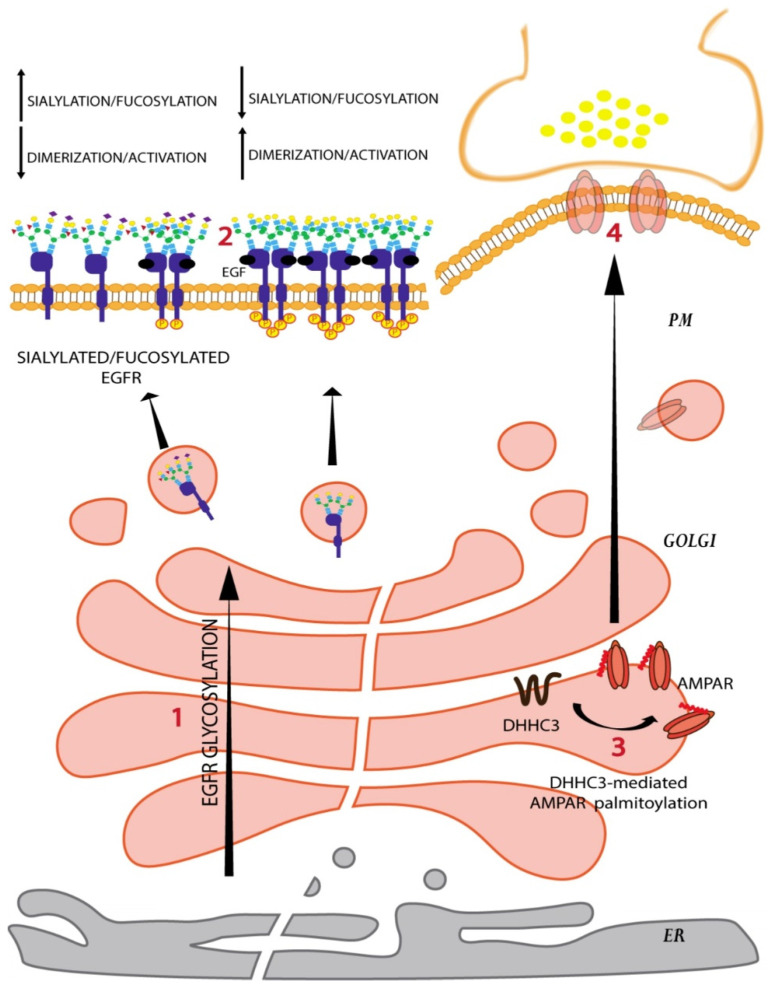
Golgi glycosylation and palmitoylation affect protein localization and function processing reactions in the Golgi,= determine the sialylation and fucosylation levels of EGFR. Increased sialylation or fucosylation reduces EGFR dimerization and activation (**1**) while reduced sialylation or fucosyaltion promotes it (**2**). The glutamate-gated ion channels AMPARs are made by different combinations of four subunits, GLUA1-4, each of which is palmitoylated. Palmitoylation of GLUA1 and GLUA2 is mediated by the Golgi resident palmitoyltransferase DHHC3 (**3**) and causes AMPAR retention in the Golgi and reduction of its expression at PM (**4**). This in turn impacts synaptic plasticity.

**Figure 5 cells-11-00368-f005:**
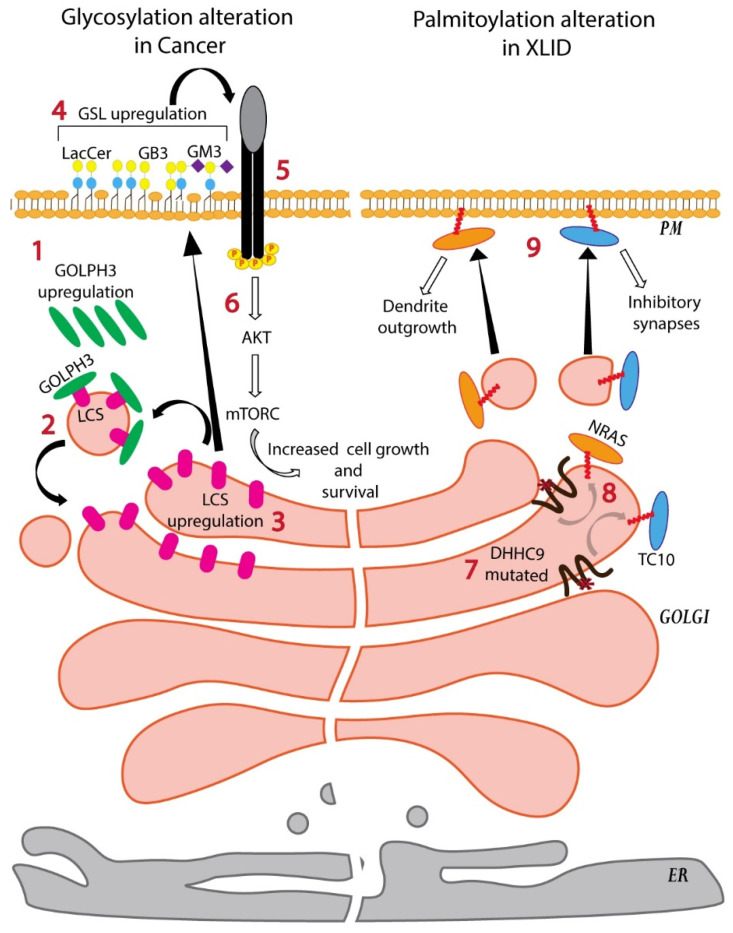
Impaired Golgi processing results in pathological conditions. The peripheral membrane protein GOLPH3 is upregulated in several solid tumours and promotes mitogenic signaling and cell proliferation (**1**). GOLPH3 binds LCS and a subset of other GSL biosynthetic enzymes, and promotes their entry into COPI vesicles for intra-Golgi retrograde trafficking and thus preventing their transport to lysosomes (**2**). This increases the levels of these enzymes in the Golgi thus promoting the biosynthesis of GSLs (**3**). Increased GSLs in the PM (**4**) likely increase the activity receptors in the PM (**5**) and that lead to increase mTOR signaling in the cell and increased cell growth (**6**). The palmitoyltransferase DHHC9 controls both dendrite outgrowth and inhibitory synapses in neurons and therefore its activity is required to maintain a balance between excitatory and inhibitory synapses. It acts by palmitoylating the GTPases NRAS and TC10, respectively, and localizing them to PM. Loss-of-functions mutations in DHHC9 found in patients with X-linked intellectual disability (XLID) (**7**) impairs the palmitoylation of NRAS and TC10 (**8**) and likely affects their PM localization (**9**). This impairment likely contributes to XLID and epilepsy.

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
