# Peer review of "Golgi Apparatus Regulates Plasma Membrane Composition and Function"

_cells, 2022, doi:10.3390/cells11030368_

Round 1
Reviewer 1 Report
This review presents a summary regarding the role of the Golgi apparatus in determining the composition and function of the plasma membrane. I do not have any special major comments or criticisms concerning the authors choice of sub-topics or the overall organization of the paper. Unfortunately, however, the text contains many mistakes, poor formulations and sentences where the meaning is difficult to follow, and which should be rephrased. Below, I point out many such places in (the beginning of) the text and come up with a list of suggestions for improvement. However, my list deals with only part of the problems and the authors - prior to publication - should subject their paper to comprehensive editing of both language and style.
Specific comments.
- PM should be spelled out in the title
- Line 12: "cell physiology and development"
- Line 21: The sentence (The PM...) should be rephrased
- Line 29: ...the appropriate membrane domain.
- Line 36: "fatty acids" rather than "lipids"
- Line 62: ...discuss three distinct processes"
- Line 65: ...appropriate domain of the PM?
- Line 96: galactosyltransferases
- Line 101: glycosaminoglycan and glycosphingolipid pathways
- Line 104: reactions happen
- Line 136: amino sugar - rather than "glycan"?
- Line 137: and as a result
- Line 139: C-terminal
- Line 152: The sentence should be rephrased
- Line 165: Reformulation required
- Line 169: Separate subtitle for S-palmitoylation/acylation?
- Line 183: Sentence does not make sense.
- Line 185: Rephrase the sentence (S-palmitoylation being...)
- Line 189: ...a variety of modified gene products...?
- Line 193: ...characteristic protein profile.
- Line 201: Rephrase the sentence (So, in...)
- Line 211: Rephrase the sentence (These...)
- Line 220: ..have a domain that specifically recognizes...
- Line 224. ...that are...
- Line 228: ...to eliminate...
- Line 247: ...happen...
- Line: 251: Rephrase the sentence (There are...)
- Line 265: ...are recognized...
- Line 312: carriers
- Line 317: Rephrase the sentence (Similar but...)
- Line 329: Rephrase the sentence (While in...)
- Line 340: its half-life
- (And so on).
Reviewer 2 Report
This is a very well written review that highlights important aspects of the Golgi function. I have only minor suggestions for improvement.
Title:
The authors should consider to change the title of the review - in my opinion it would be more appropriate to replace “activity” with “function”.
Line 94: N-glycosylation is initiated by the oligosaccharyltransferase which is not a typical glycosyltransferase – please rephrase.
Line 107: asparagine instead of Asparagine.
Line 132: O-glycan instead of o-glycan.
Line 139: C-terminal.
Line 142: glucuronic acid
Line 222: please define StAR
Line 224: that “are” involved.
Line 240 – Figure 2: the illustration shows the TGN and ER, but not the Golgi. Therefore I would not call it at the Golgi apparatus.
Line 247: are there capping reactions in the TGN? Typically terminal sugars (e.g. sialic acid) are transferred in the trans-Golgi, while the TGN is biosynthetically inactive and a sorting compartment.
Line 361: is known to have.
Line 362: “12 N-glycans that are processed”. The sites are glycosylated and the glycans are processed.
Line 369: determine
Line 480: zebrafish
Lines 548 and 549: GOLPH3 instead of Golph3.
Line 559: associated with several…
